# DISENTANGLING THE ROLES OF REPRESENTATION AND SELECTION IN DATA PRUNING (FOR FINE-TUNING)

## ABSTRACT

Data pruning, the process of carefully selecting a small subset of training data, has been shown to improve both training efficiency and performance. It typically involves two steps: (1) obtaining a representation for each instance, and (2) applying a selection algorithm using these representations. However, the distinct roles of these two steps, as well as their interactions, remain unclear. To address this, we conduct a systematic study of data pruning, focusing on NLP fine-tuning. Our theoretical and empirical findings reveal that data representation often plays a more fundamental role than the selection algorithm: gradients, despite being computationally expensive, provide stronger pruning signals than other representations, making gradient-based methods consistently outperform cheaper alternatives. We also demonstrate that different selection algorithms excel in specific scenarios but are heavily influenced by the chosen representation. These insights provide clear guidelines for future research and practical applications.

## 1 INTRODUCTION

The remarkable success of modern deep learning is largely driven by vast training datasets (Kaplan et al., 2020; Hoffmann et al., 2022; Sardana et al., 2024). However, scaling the size of datasets comes with great computational costs. Fortunately, recent studies have shown that by carefully selecting a small subset of the original large dataset, a process known as *data pruning*, it is possible to improve training efficiency while also improving generalization performance (Paul et al., 2021; Sorscher et al., 2022; Du et al., 2023; Xia et al., 2024). Moreover, effective pruning methods can offer insights into learning algorithms and the roles of training data (Koh & Liang, 2017; Ilyas et al., 2022).

Most data pruning methods involve two steps: First, obtaining a *representation* for each instance in the original training set (e.g., hidden states from a pretrained language model); second, selecting instances based on these representations given a *data budget* (e.g., 30% of the training set), according to a *selection algorithm*. Despite the success of data pruning, *existing studies treat these two steps as a single process*, leaving fundamental questions about the roles of representations and selection algorithms unanswered: *Which data representations and selection algorithms are most effective, and what is their suitability for different tasks? How do different representations impact the data points selected by each selection algorithm?* To address these questions, we conduct a thorough study of data pruning through both theoretical and empirical lenses, focusing on NLP fine-tuning tasks. Our contributions are as follows:

1. We offer a systematic overview of existing data pruning methods (§2). From this overview, we identify three common ways to create representations: based on *training dynamics*, *hidden states*, and *gradients*. We also identify three common objectives of selection algorithms: maximizing *difficulty*, maximizing *diversity*, and maximizing *validation performance*.

2. We both theoretically analyze the pruning signals contained in different representations (§3), and empirically validate that the representations of stronger signals, despite being computationally more expensive, are more effective (§4). Specifically, we find that *data pruning based on gradients often performs the best*. In contrast, data pruning based on hidden states, which is computationally cheaper, often does not perform better than random selection, especially with low data budgets. Our experiments evaluate six representative methods across common representations and selection algorithms, covering a wide range of

NLP tasks: hate speech detection (classification), commonsense reasoning (multiple choice), and summarization (text generation).

3. Our empirical experiments (§4) further show that maximizing difficulty requires a higher data budget to perform well, maximizing validation performance excels when train-test distributions differ, and that maximizing diversity in general does not work very well.

4. Through both interpretable experiments with synthetic data (§3) and common NLP tasks (§4), we show that *with the same selection objective, different data representations can lead to drastically different selections of instances*. For instance, when the aim is to select the most difficult instances, using hidden states results in selecting instances far from the decision boundary, while using gradients select instances close to it. Moreover, we show that *the sensitivity of selection algorithms towards the change of representations vary*, and that *data representations often matter more than selection algorithms*.

Our findings highlight the strengths and limitations of current data pruning methods, offering clear guidelines for selecting appropriate representations and algorithms given specific tasks and constraints. We recommend prioritizing the development of efficient gradient-based methods, due to their superior performance and better interpretability (Pruthi et al., 2020; Park et al., 2023).

## 2 OVERVIEW OF DATA PRUNING METHODS

Various data pruning methods have been proposed. They typically follow two steps for data selection: (1) obtaining a representation for each instance, and (2) using an algorithm to select instances based on their representations. Regarding representations, most methods rely on one of the three types: training dynamics, hidden states, and gradients. Our overview is structured accordingly. While recent studies have explored external large language models, by prompting and distillation, particularly in instruction tuning (Sachdeva et al., 2024; Chen et al., 2024; Lu et al., 2024; Liu et al., 2024), *we focus on representations from the model that we are training*. This allows us to analyze signals that directly reflect its learning behavior.

Regarding selection algorithms, most methods are built on one or more of the following objectives: retaining the most difficult data instances, maximizing the diversity of selected data, and retaining the training instances that improve a model's performance on held-out data the most.

### 2.1 TRAINING-DYNAMIC-BASED METHODS

Most data pruning methods based on training dynamics aim to maximize the ratio of difficult instances being selected, where the difficulty is defined by heuristics like low prediction confidence and high training loss. For example, Toneva et al. (2019) determine the difficulty of examples by the earliest epoch after which that example is always correctly classified; Jiang et al. (2019) keep the examples with the highest training loss; Swayamdipta et al. (2020) and Du et al. (2023) use the standard deviation and mean of prediction probabilities of the correct class across different epochs; Paul et al. (2021) keep the examples with the largest error norm; Baldock et al. (2021) select examples that need to pass through more layers before being correctly classified; and Marion et al. (2023) and Kwok et al. (2024) quantify difficulty by perplexity. Moreover, Maini et al. (2022) show that examples that are forgotten slower when training on a held-out subset are more rare and thus worth keeping. In the context of instruction tuning, Li et al. (2024) quantify the difficulty of a response by the ratio of its losses between generations with and without its instruction.

Other selection objectives have also been explored. For instance, Mindermann et al. (2022) prioritize training on learnable, worth-learning, and not-yet-learned examples, quantified by the loss difference between the model itself, and a small reference model trained on a held-out dataset. Furthermore, Yang et al. (2024) aim to diversify the training data by clustering instances based on their training losses across different epochs and sampling evenly from each cluster.

### 2.2 HIDDEN-STATE-BASED METHODS

Hidden-state-based methods often exploit similarities between data samples to realize various selection objectives. For example, Sorscher et al. (2022) first perform a $k$-means clustering of hidden

states, then select the ones that lie farther from their cluster centroids, because these instances are less prototypical and therefore more likely to be difficult. Abbas et al. (2023) and Tirumala et al. (2023) extend this approach by identifying and removing semantic duplication after clustering. This de-duplication step helps balance data diversity and difficulty by discarding redundant instances.

## 2.3 GRADIENT-BASED METHODS

Another line of methods uses gradient information of each training instance to estimate its importance. Specifically, most gradient-based methods estimate the impact of removing instances by simulating re-training scenarios. These approaches use gradient-based tools like influence functions (Koh & Liang, 2017; Pruthi et al., 2020) and datamodels (Ilyas et al., 2022; Park et al., 2023).

The most common objective is to maximize or maintain validation performance. For example, Xia et al. (2024) and Engstrom et al. (2024) discard training instances with low contribution towards the validation performance of the target task. They approximate this influence by using the TracIn influence function (Pruthi et al., 2020) and datamodels (Ilyas et al., 2022). Moreover, Killamsetty et al. (2021) and Yang et al. (2023) keep instances that likely result in similar models as when training on the full dataset.

Gradients are also often used to select difficult instances. For example, Feldman & Zhang (2020) demonstrate that data instances of high self-influence (i.e., training on itself contributes more to its prediction) are more difficult and help generalization. Using the TracIn influence function, this self-influence score can be estimated as the gradient norm. Similarly, Thakkar et al. (2023) prioritizes data instances of different self-influences at different stages of pretraining, to limit the influence of noisy data while focusing on higher-quality ones; and Bejan et al. (2023) uses automatic curriculum learning to filter noisy data.

## 3 A TALE OF DATA REPRESENTATIONS AND SELECTION ALGORITHMS

This section studies the distinct roles of data representations, selection algorithms, and their interactions. Specifically, we focus on six representative methods that span all three major types of data representations and selection objectives. First, we provide in-depth explanations of each method (§3.1). Next, we present a theoretical analysis of the signals each representation offers (§3.2). Finally, we conduct interpretable synthetic experiments to study how the data instances favored by each selection algorithm are shaped by the chosen data representations (§3.3).

**Notation** We now summarize the notation used in this paper. We denote the original training set with $N$ instances as $\mathcal{D} = \{(x_i, y_i)\}_{i=1}^{N}$. The selected subset of data is represented by $\mathcal{S} \subset \mathcal{D}$. We use $B$ to denote the data budget (e.g., $B = |\mathcal{S}| = 0.2N$). For a data point $(x_i, y_i)$ and a model with parameters $\theta$, we use $f_\theta(x_i)$ to denote the model's logits, $\ell(f_\theta(x_i), y_i)$ to denote the loss, and $p_\theta(y_i|x_i)$ to denote the prediction probability of the *correct class* or token. Moreover, we use $h_\theta(x_i)$ to denote the last layer hidden state of input $x_i$, and $g_\theta(x_i, y_i) = \nabla_\theta \ell(f_\theta(x_i), y_i)$ to represent the gradient. Moreover, we train each model for $T$ epochs, use $p_t(y_i|x_i)$ to denote the prediction probability of the correct class at epoch $t \leq T$, and use $\eta_t$ to denote the learning rate at epoch $t$.

### 3.1 PRELIMINARIES: DETAILS OF DATA PRUNING METHODS

Data pruning typically involves two steps under a given data budget (e.g., 20% of the original dataset). First, a *reference model* is used to obtain representations for each instance, such as hidden states from a pretrained model. Second, a selection algorithm chooses a subset based on these representations, e.g., the ones that are farthest from the clustering centroids, following a selection objective, e.g., selecting the most difficult instances. The selected instances are then used to a train new *main model* which is the final model of interest. We summarize the representations and selection objectives of these methods in Appendix B.

We first describe two training-dynamic-based methods, **Hard-to-Learn** and **SmallToLarge** (S2L). These training dynamics (e.g., training losses) are collected from a reference training run, during which a reference model is trained on the original dataset.

**Hard-to-Learn**   The Hard-to-Learn method is based on a simple intuition: training instances that are **difficult** for models to fit often contain fewer regularities, and these instances can improve model generalization (Swayamdipta et al., 2020; Jiang et al., 2021). Concretely, it represents each training instance by a score computed from the **training dynamics** of a reference model. In classification tasks, for a given instance $(x_i, y_i)$, its score is defined as *the average prediction probability of the correct label across different epochs*, i.e., $s_{\text{hard}}(x_i, y_i) = \frac{1}{T} \sum_{t=1}^{T} p_t(y_i|x_i)$. Instances with the lowest scores are selected for training the main model, i.e., $\mathcal{S} = \arg\min_{|\mathcal{S}|=B} \sum_{(x_i,y_i)\in\mathcal{S}} s_{\text{hard}}(x_i, y_i)$. To extend this concept to generation tasks, Bhatnagar et al. (2022) and İnce et al. (2023) replace $s_{\text{hard}}$ with the inverse perplexity. Additionally, Jiang et al. (2021) show that $s_{\text{hard}}$ correlates well with the expected accuracy of an instance when it is excluded from the training data. This correlation also suggests that Hard-to-Learn instances contain fewer regularities, supporting the intuition.

**SmallToLarge**   Also based on **training dynamics**, SmallToLarge (S2L) aims to select **diverse** instances. Specifically, Yang et al. (2024) observe that training instances with different loss trajectories across epochs likely require different knowledge. To make sure different skills are evenly represented in the subset $\mathcal{S}$, S2L performs three steps. Initially, it represents each training instance by its *cross entropy loss trajectory*. Then, it performs a $k$-means clustering on these trajectories, and sorts these clusters by size in an ascending order. Finally, it iteratively samples $(B - |S|)/(K - k + 1)$ instances from each cluster[1], where $K$ is the number of clusters, and $k$ is the current cluster index. This sample size choice helps S2L prioritize smaller clusters to increase the diversity of the selected data.

Next, we discuss two hidden-state-based methods, **Prototypicality** and **SemDeDup**. Hidden states are usually computed using a reference pre-trained model, and thus require no further training, making them computationally more efficient than other methods.

**Prototypicality**   The Prototypicality method (Sorscher et al., 2022) aims to select **difficult** instances by exploiting their *similarities*: it measures difficulty based on how *prototypical* an instance is in the dataset. Specifically, after representing instances by their **hidden states**, prototypicality applies $k$-means clustering and ranks instances based on their distances to their cluster centroids. Instances with larger distances are considered less prototypical, more difficult, and selected for further training.

**SemDeDup**   Building on Prototypicality, SemDeDup adds one more step to account for data **diversity** besides **difficulty** (Abbas et al., 2023). Concretely, after the clustering step of Prototypicality, SemDeDup identifies semantically duplicate pairs of instances within each cluster using cosine similarities of their **hidden states**. For each identified duplicate pair, it retains the instance that lies farther from the cluster centroid, thereby prioritizing diversity while maintaining difficulty.

Finally, we discuss two gradient-based methods, **LESS** and **Memorization**. They require training a reference model on the original dataset, and computing gradients using different checkpoints. This makes them very costly, because they require performing back-propagation with batch sizes of 1 to obtain the gradients, which have a high dimensionality, equal to the number of model parameters.

**LESS**   Unlike the above methods that only rely on representations of the training data, LESS assumes the availability of a validation set. It aims to select instances that maximize the **validation performance**, measured by *validation loss reduction* (Xia et al., 2024). Naively, this estimation requires retraining models on a large number of random training subsets, which is prohibitively expensive for modern neural models. Therefore, LESS employs influence functions for approximation (Pruthi et al., 2020). Specifically, given the $t$-th reference model checkpoint $\theta_t$, LESS estimates the loss reduction of a training instance $(x_i, y_i)$ w.r.t. a validation instance $(x_{\text{val}}, y_{\text{val}})$ by the dot product of their (normalized) **gradients**. For multiple checkpoints, LESS performs weighted averaging by learning rates. Formally, $s_{\text{LESS}}(x_i, y_i) = \sum_{t=1}^{T} \eta_t g_{\theta_t}(x_i, y_i) \cdot g_{\theta_t}(x_{\text{val}}, y_{\text{val}})$. The instances with the highest scores are kept. Moreover, for efficiency, LESS uses LoRA (Hu et al., 2022) and random projection (Park et al., 2023; Johnson & Lindenstrauss, 1984) for dimensionality reduction. Additionally, LESS considers optimizer states when computing training gradients.

**Memorization**   Feldman & Zhang (2020) define memorization in training as the loss increase of a training instance before and after it is removed from the training set, i.e., self-influence. Following a

---

[1]For simplicity, here we reload $|S|$ as the number of already sampled instances.

similar intuition as Hard-to-Learn, Feldman (2020) argues that instances with high memorization scores are usually more **difficult** and thus contribute more to generalization. Similar to LESS, memorization is also usually approximated by influence functions using **gradients** (Thakkar et al., 2023; Bejan et al., 2023; Pruthi et al., 2020). In this work, following Pruthi et al. (2020), we compute the memorization score of a training instance $(x_i, y_i)$ by $\sum_{t=1}^{T} \eta_t g_{\theta_t}(x_i, y_i) \cdot g_{\theta_t}(x_i, y_i)$. Inspired by Xia et al. (2024), we also adopt LoRA and random projection for efficiency.

## 3.2 What do different representations reveal?

Data representation is central to data pruning, yet it remains unclear *how different representations vary in the signals they retain*. This section offers theoretical insights into this question. Specifically, as described in §3.1, most data pruning methods rely on the similarity between instances based on their representations (e.g., S2L, Prototypicality, and LESS). We therefore ask: *How does the notion of similarity between two instances change across different representation spaces?* Our analyses build insights into the pruning signals from different representations, which we will further study in the following synthetic (§3.3) and NLP task (§4) experiments.

For clarity of analysis, we consider a binary classification task with labels $y \in \{-1, +1\}$, optimized with binary cross-entropy loss. For the representations, we focus on the prediction probability of the correct class, the hidden states before the classification layer (parameterized by $w$, i.e., the classifier or language modeling head), and the gradients of this classification layer.[2] Given a training instance $(x_i, y_i)$ and a model $\theta$, let $h(x_i; \theta)$ be the hidden states. We can derive the prediction probability of the correct class $p_\theta(y_i|x_i) = \sigma(y_i w^T h(x_i; \theta))$, where $\sigma$ is the sigmoid function. We can also compute the gradients as $g_w(x_i, y_i) = \nabla_w \ell(f_\theta(x_i), y_i) = (1 - \sigma(y_i w^T h(x_i; \theta))) y_i h(x_i; \theta)$. Specifically, we use Euclidean distance as the similarity metric, and consider hidden states as the fundamental units, given their involvement in both prediction probability and gradient expressions.

We first consider the correct class prediction probability difference. Formally, given two training instances $(x_i, y_i)$ and $(x_j, y_j)$, this is $|p_\theta(y_i|x_i) - p_\theta(y_j|x_j)| = |\sigma(y_i w^T h(x_i)) - \sigma(y_j w^T h(x_j))|$. Because the Sigmoid function is smooth and monotonically increasing, when this difference is small, $w^T |y_i h(x_i) - y_j h(x_j)|$ should also be small. We can identify two cases: when the labels agree, i.e., $y_i = y_j$, $h(x_i)$ and $h(x_j)$ should be close in both direction and length (i.e., L2 norm) *after being projected onto $w$*; when labels do not agree, i.e., $y_i \neq y_j$, $h(x_i)$ and $h(x_j)$ should be opposite but of similar length. In other words, in contrast to distances between hidden states, i.e., $\|h(x_i) - h(x_j)\|$, the difference in prediction probabilities also considers label agreement.

Let us now analyze the Euclidean distance between instances represented as gradients[3],

$$\|g_w(x_i, y_i) - g_w(x_j, y_j)\|_2 = \sqrt{\|g_w(x_i, y_i)\|_2^2 + \|g_w(x_j, y_j)\|_2^2 - 2g_w(x_i, y_i)^T g_w(x_j, y_j)}$$

$$= \sqrt{\text{err}_i^2 \|h_i\|_2^2 + \text{err}_j^2 \|h_j\|_2^2 - 2\text{err}_i \text{err}_j h(x_i)^T h(x_j) y_i y_j}, \quad \text{where } \text{err}_i = 1 - \sigma(y_i w^T h(x_i)).$$

Here $\text{err}_i$ is the prediction error of instance $(x_i, y_i)$. We discuss two scenarios. First, if the gradients are normalized (as in LESS), the first two terms above are both 1. The third term depends on the cosine similarity (equivalent to the dot product) of hidden states and label agreement. To minimize gradient distance, when $y_i = y_j$, $h(x_i)$ and $h(x_j)$ should have high cosine similarities; when $y_i \neq y_j$, $h(x_i)$ and $h(x_j)$ should be of low cosine similarities. This is similar to the prediction probability case, except that *the similarity between hidden states is independent of both their lengths and $w$*.

Second, when gradients are not normalized, the gradient distance is $\|\text{err}_i h(x_i) - \text{err}_j h(x_j)\|_2$ and $\|\text{err}_i h(x_i) + \text{err}_j h(x_j)\|_2$ respectively when $y_i = y_j$ and $y_i \neq y_j$. Similar to the case for prediction probability, when the gradient distance is small, hidden states should be similar when their labels agree, and dissimilar when their labels disagree. However, here the hidden states are scaled by their prediction errors: if we assume that the length of hidden states are similar (which holds in practice), we require their prediction errors to be similar as well.

In summary, similarities based on training dynamics and gradients are closely related to hidden states. Besides, they both implicitly encode *label agreement*, and gradients also encode *prediction errors*.

---

[2]Classification layer gradient is a popular choice for large-scale models, e.g., Pruthi et al. (2020).

[3]Our conclusions apply to cosine similarity, as used in LESS (equates to using normalized gradients here).

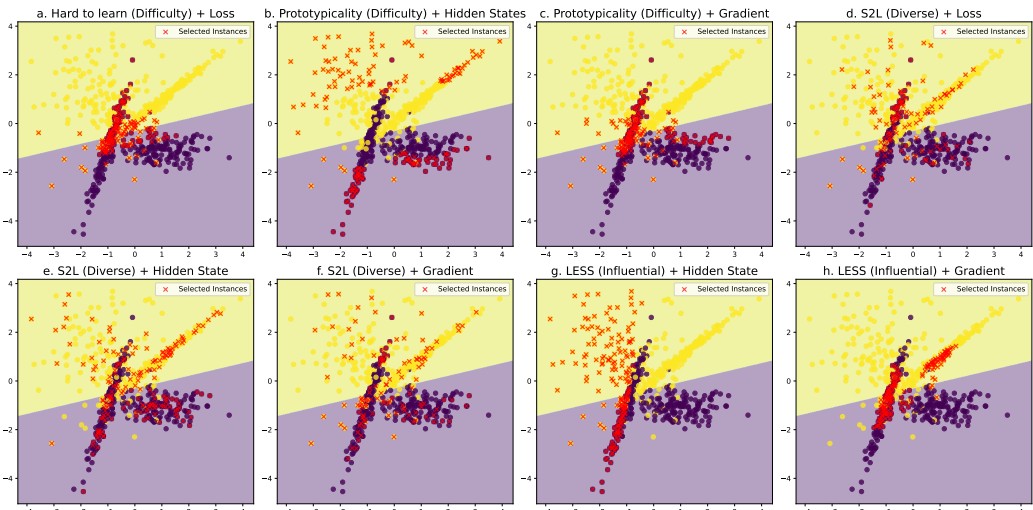

Figure 1: Interactions between data representations and selection algorithms. We generate 600 data points from a 2D Gaussian mixture model and compare different methods to select 30% (180) of the data points. The color represents the ground-truth label, and the red Xs are the selected data points. (1) Using different representations with the same selection objective or algorithm leads to drastically different subset selections (a–c, g–h); (2) The sensitivity of selection algorithms towards the used representations varies (e.g., compare d–f with b–c).

### 3.3 HOW DO REPRESENTATIONS AND SELECTION ALGORITHMS INTERACT?

This section analyzes how different representations affect the outcomes of selection algorithms. Specifically, we aim to answer two questions: *does a given selection algorithm choose different subset selections when using different representations? Which selection algorithms are more sensitive to representation changes?* For this purpose, we first conduct an interpretable analysis using a synthetic dataset. Building on these insights, we will empirically study more complex scenarios in §4.

To compare selection objectives across different representations, we focus on three selection algorithms that are representation-agnostic: prototypicality (prioritizing difficulty), S2L (prioritizing diversity), and LESS (prioritizing influence on validation set performance). We also include Hard-to-Learn for comparison. We generate 600 data points from a 2D Gaussian mixture model and use each method to select 30% (180) of the data points (Figures 1 and 6). We train a logistic regression classifier to serve as the reference model to obtain training dynamics and gradients, and the original data points are used as hidden states. Our observations are as follows.

First, *even when data pruning methods have the same objective, the representations and selection algorithms used can result in drastically different subset selections*. For example, both Hard-to-Learn and Prototypicality aim to select difficult instances. To achieve this, Hard-to-Learn selects instances with low correct class probabilities, which favors the ones that are located near the decision boundary (Figure 1a). In contrast, Prototypicality clusters data instances based on hidden states, and selects those that are far from the cluster centroids. As a result, it selects instances from the sparsely populated regions (Figure 1b).

Second, *even the same selection algorithm can prefer different data points*, when using different representations. As discussed in §3.2, instances with similar hidden states but different labels are far apart in gradient space (§3.2). For example, when using gradients, Prototypicality favors instances near the decision boundary as they are far from centroids (Figure 1c). In contrast, with hidden states, it selects instances far from the decision boundary, as mentioned above (Figure 1b).

Third, *selection algorithms vary in sensitivity towards representations*. For example, S2L selects similar instances with different representations (Figures 1d–f). Indeed, diversity-preserving algorithms are less affected by the choice of representation because they sample evenly from different regions. In contrast, both Prototypicality (Figure 1b–c) and LESS (Figure 1g–h) select remarkably different

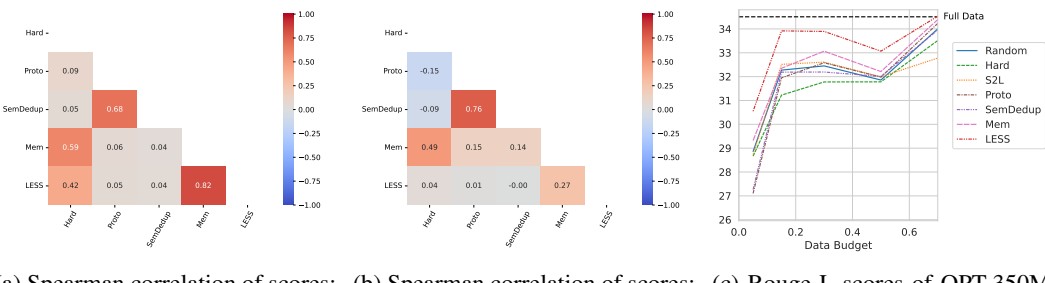

(a) Spearman correlation of scores: DeBERTaV3$_{Large}$ on WinoGrande

(b) Spearman correlation of scores: OPT-350M on DialogSum

(c) Rouge-L scores of OPT-350M on DialogSum

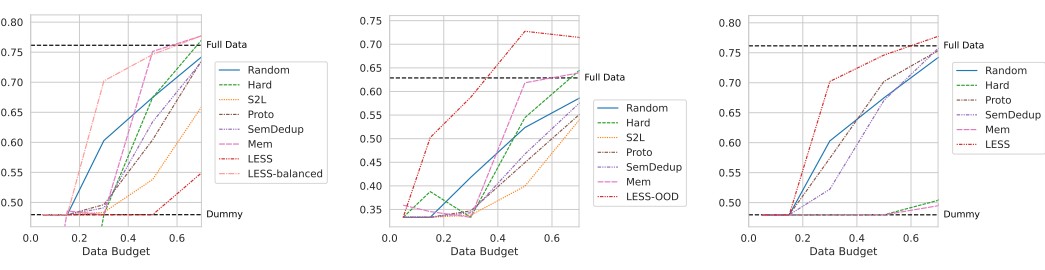

(d) F1 scores of DeBERTaV3$_{Large}$ on CAD

(e) F1 scores of DeBERTaV3$_{Large}$ on DynaHate

(f) F1 scores of DeBERTaV3$_{Large}$ on CAD with label matching

Figure 2: Spearman correlation of scores calculated by different methods (2a-2b, all methods select instances with the highest scores to retrain) and their model performance under different data budgets (2c-2f). Here Hard, Proto, and Mem refer to Hard-to-Learn, Prototypicality, and Memorization. LESS-OOD refers to LESS using DynaHate as the validation set. We also experimented with forced label matching. Regarding correlations, we identify two pairs of methods with moderate to high correlations: Hard-to-Learn and Memorization, and Prototypicality and SemDedup. Regarding model performance, we show that 1) *data pruning is not always effective*: hidden-state-based methods do not outperform random selection, while LESS (with label matching) is competitive on all tasks; 2) *different objectives suit different scenarios*: maximizing difficulty performs well with high data budgets, and prioritizing validation performance works better when there is a mismatch between the train and test distributions; preserving diversity works less well.

data points using hidden states and gradients, because instance similarities computed by gradients additionally encode their label agreement and the magnitudes of their prediction errors (§3.2).

## 4 EMPIRICAL EVALUATIONS

### 4.1 EXPERIMENTAL SETUP

Building on our theoretical and synthetic analyses (§3), we now study the impact of various data representations and selection algorithms from §3.1 on NLP tasks. We evaluate their consistency (§4.2) and performance across different data budgets (§4.3). We also perform ablation studies to examine the role of each component, as well as their effectiveness in handling label noise (§4.4).

We evaluate on three diverse NLP tasks: CAD (hate speech detection, binary classification (Vidgen et al., 2021)), WinoGrande (common sense reasoning, multiple choice (Sakaguchi et al., 2021)), and DialogSum (summarization, generation (Chen et al., 2021)). For CAD, we include Dyna-Hate (Kiela et al., 2021) as an OOD test set. For CAD and WinoGrande we use DeBERTaV3$_{Large}$ and DeBERTaV3$_{Base}$ (He et al., 2023), and for DialogSum we use OPT-125M and OPT-350M (Zhang et al., 2022).[4] We experiment with six data budgets: 5%, 15%, 30%, 50%, 70%, and 100% of the original dataset, and fine-tune all models for 15 epochs. For fair comparisons, we use the same

---

[4]We use relatively small models to avoid huge computation during both training (trained 1200+ models for controlled comparisons) and gradient projection (can take > 10 times because of the high dimensionality).

reference models as the main models (more details in Appendix A), although recent works (Du et al., 2023; Yang et al., 2024) have shown the promising performance of using efficient reference models. We also experimented with forced label matching to match the original dataset (LESS-balanced in Figure 2d, and Figure 2f), to address highly skewed pruned datasets (details in §4.3).

## 4.2 CONSISTENCY BETWEEN DATA PRUNING METHODS

This section analyzes whether different pruning methods rank data points similarly. We use Spearman's $r$ to compare the scores assigned to each instance by various methods[5] because (1) rank-based correlation are more suitable for comparing scores of different ranges, and (2) it allows for evaluation without setting a fixed data budget. Results for DeBERTaV3$_{Large}$ on WinoGrande and OPT-350M on DialogSum are shown in Figures 2a and 2b, with additional results in Appendix C.2.

Consistent with §3.3, we observe that *the same objective can lead to different data selections*: For instance, although both Prototypicality and Hard-to-Learn aim to select difficult instances, they show almost no correlation. However, the pairs *Prototypicality - SemDedup* and *Hard-to-Learn - Memorization* show moderate consistency across datasets and models. This aligns with our expectations: Both Prototypicality and SemDedup are based on clustering hidden states, with SemDedup adding a step for semantic deduplication. Meanwhile, both Hard-to-Learn and Memorization select difficult instances whose predictions are barely improved by training on other instances (Jiang et al., 2021). Since Memorization requires gradients, it is far more costly, making Hard-to-Learn the more scalable option, although the moderate correlation suggests there are still differences in their data selections.

## 4.3 PERFORMANCE UNDER DIFFERENT DATA BUDGETS

Our previous analyses focus on how different components influence data selection. This section further analyzes their implications on model performance under different data budgets, to answer questions regarding *which data pruning method to use in specific scenarios*. Three baselines are considered: random selection (Rand), the full original dataset (100% data budget), and a dummy predictor (Dummy, the better one between a randomized predictor and a majority class predictor), as performance below it is considered as failed (Mosbach et al., 2021). We make three observations (Figures 2c to 2f, and Appendix C.3).

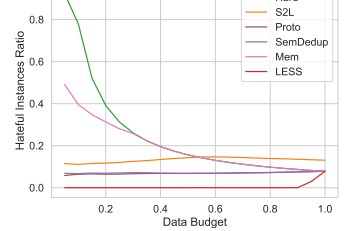

First, *data pruning is not always effective*. For example, Hard-to-Learn and S2L perform underperform random selection on DialogSum (Figure 2c) and CAD (Figure 2d). Surprisingly, hidden-state-based methods (Prototypicality and SemDedup), perform worse or similar to random selection on all tasks, suggesting clustering pretrained hidden states is not suitable for our setting.[6]

Figure 3: DeBERTaV3$_{Large}$ on CAD: the ratio of hateful instances in selected instances, under different data budgets.

Second, *higher data budgets are needed for methods that select difficult instances*, i.e., Hard-to-Learn and Memorization. They achieve good performance with $> 30\%$ data budgets on CAD (Figure 2d) and WinoGrande (Figure 8e), but they both struggle with lower data budgets ($< 30\%$) (e.g., worse or comparable to Dummy). This is consistent with Swayamdipta et al. (2020): including only the most difficult instances will make models fail to converge. Moreover, due to their consistency in both scores (§4.2) and classification performance, future studies may prioritize Hard-to-Learn over Memorization for selecting difficult instances in classification tasks, aiming for better efficiency.

Third, gradient-based methods (LESS and Memorization) are competitive (Figure 2c-2e), indicating that *gradients are effective data representations*. Between them, *LESS performs better but requires label matching for highly skewed datasets*. For example, on CAD, where 90% of instances are non-hateful, directly applying LESS yields poor results (Figure 2d).

---

[5]All methods, except S2L, assign scores to instances and retain the highest-scoring ones. Since S2L only assigns cluster labels, we exclude it from this analysis. We also experimented with overlaps under different data budgets and obtain consistent observations.

[6]Note that our experiments differ from previous studies that use hidden states, as they focused on high data budget settings (Sorscher et al., 2022) and noisy pretraining datasets (Abbas et al., 2023).

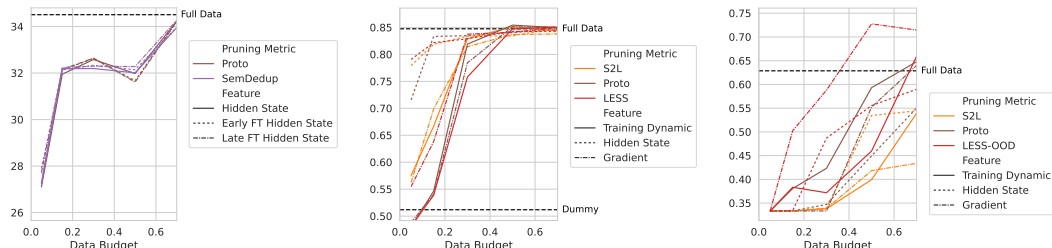

(a) Pretrained vs. Fine-Tuned (FT) Hidden States: OPT-350M on DialogSum (Rouge-L)

(b) Representation-Selection Relation: DeBERTaV3$_{Large}$ on WinoGrande (Accuracy)

(c) Representation-Selection Relation: DeBERTaV3$_{Large}$ on DynaHate (F1)

Figure 4: Ablation studies on pretrained vs. fine-tuned hidden states (4a), and using different data representations with the same selection algorithm (4b-4c).

We hypothesize the reason to be as follows. LESS ranks training instances based on their gradient similarity to validation data. However, gradient similarities depend on labels: since hidden states usually have $> 0$ cosine similarities (known as anisotropic space), instances with the same label have higher gradient similarities (§3.2). This makes LESS over-select instances with the majority label when using unbalanced validation sets, e.g., non-hateful instances in CAD, leading to pruned datasets with even more skewed label distributions.

We validate this by plotting the ratio of hateful labels in selected instances across different methods and data budgets in Figure 3. We confirm that non-hateful instances have much higher LESS scores than hateful ones, with very few hateful instances being selected until the data budget reaches 90%, unlike other methods.[7] To address this, we enforced the selected label ratio to match the original dataset, which substantially improves LESS performance (Figure 2f). However, this constraint can harm methods like Hard-to-Learn and memorization, which favor hateful instances.

### 4.4 ABLATION STUDIES

This section presents ablation studies to explore the impact of key components, including fine-tuned versus pretrained hidden states, the interaction between representations and selection algorithms (as in §3.3 but focusing on task performance), and the ability of these methods to handle label noises. We include our results in Figure 4 and Appendix C.4.

**Fine-Tuned Hidden States** Our previous results show that hidden-state-based methods perform comparably to random selection. However, fine-tuning could potentially encode task-specific and label information into hidden states, helping identify prototypical and diverse training instances.

We therefore study whether using fine-tuned hidden states can improve model performance. Specifically, we experiment with two types of fine-tuned hidden states: early (fine-tuned for one epoch, retaining more pretrained knowledge) and late (fine-tuned for 15 epochs, encoding more task-specific and label information). The results of both prototypicality and SemDedup are shown in Figure 4a. *Fine-tuned hidden states can barely improve model performance*: there is little difference between using different hidden states, and they all perform comparably to random selection.

**Representation-Selection Relation** It remains unclear whether the effectiveness of different data pruning methods stems primarily from the representations they use, the selection algorithms they use, or a specific combination of both. Similar to §3.3, to disentangle the contributions from these different components, we conduct a systematic study that combines all three different data representations with the selection algorithms of S2L, Prototypicality, and LESS, which respectively prioritize difficulty, diversity, and validation performance. We show the results for DeBERTaV3$_{Large}$ on WinoGrande and DynaHate in Figure 4b and Figure 4c, and make two observations.

---

[7]Sorscher et al. (2022) reported a similar imbalance for Prototypicality. We also observe marginally more skewed pruned datasets, and forced matching slightly improve their performance (especially with high data budgets, which is their original setting). However, their performance is overall comparable to random selection.

First, *similar representations often lead to similar performance.* For example, Figure 4b shows that the performance of using the same representation with different selection algorithms are more similar than using different representations with the same selection algorithm. This indicates that, *compared to selection algorithm, representation plays a more fundamental role*.

Second, *the optimal selection algorithm is task-dependent* and still plays a key role. For example, Figure 4c shows that methods using the LESS selection algorithm outperform others on DynaHate. One possible explanation is that DynaHate is an OOD test set that presents a substantial distribution shift from the training data (CAD), while the LESS selection algorithm compensates for this shift by selecting instances matching the validation set. This observation suggests that future practitioners should choose algorithms based on goals: prioritizing difficulty for better robustness, and prioritizing contribution to validation performance when adapting models to a different domain.

**Noise Detection**    An important applications of data pruning is removing noise from training data (Swayamdipta et al., 2020; Paul et al., 2021). To evaluate the noise detection capabilities of different methods, we create two noisy versions of each dataset by changing 2% and 5% of the training labels. Concretely, we flip the labels for CAD and WinoGrande (both datasets have binary labels), and substitute the summaries of DialogSum with a randomly selected one. We then train reference models on these noisy datasets, and use different methods to select data. We use fine-tuned hidden states (both early and late) instead of pretrained hidden states, as pretrained hidden states do not encode label-relevant information.

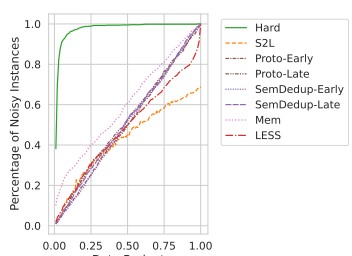

Figure 5: The percentage of selected noisy instances in all noisy instances of DeBERTaV3$_{Large}$ on WinoGrande, with 2% noise rate.

We analyze the percentage of selected noisy instances out of all noisy instances across data budgets. Figure 5 shows results for DeBERTaV3$_{Large}$ on WinoGrande (2% noise rate), with additional results in Appendix C.4.[8] Except for Hard-to-Learn, which is proposed for noise detection (Swayamdipta et al., 2020), none of the methods effectively filter out noisy instances. However, Hard-to-Learn tends to select more noisy instances, which conflicts with our goal of identifying clean data. Consequently, *none of the data pruning methods are suitable for training with noisy data.* Notably, although Memorization shows moderate consistency with Hard-to-Learn (§4.2), it only ranks noisy instances marginally higher than clean instances.

## 5    CONCLUSION

**Summary**    Despite the success of data pruning, the roles of its two key components – data representation and selection algorithm – and their interactions, are not well understood. In this work, we have conducted both theoretical and empirical analyses on these two choices in fine-tuning tasks. Our results highlight the importance of using rich data representations, showing that gradient-based methods consistently outperform computationally cheaper alternatives. Additionally, the optimal selection algorithm varies depending on specific use cases, although the outcomes are heavily influenced by the chosen representation. Moving forward, our findings stress the need for the development of **scalable supervised representations**, i.e., representations that encode label-relevant information, as more effective alternatives to the current unsupervised ones, e.g., pretrained hidden states.

**Limitations**    The most notable limitation of our work is its focus on task-specific fine-tuning that leaves multi-task instruction tuning unexplored. This is largely due to (1) the huge amount of computation required to conduct rigorous controlled studies as ours, and (2) the challenges in scalable and low-cost evaluation (Zheng et al., 2023). Future studies could explore instruction-tuning with synthetic data, which has been recently shown to be effective for proof-of-concept studies (Allen-Zhu & Li, 2024). Moreover, we focus on methods that do not require external models (e.g., prompting language models to evaluate example quality). Future work can extend our analysis to them.

---

[8]S2L samples instances from different clusters in a balanced way (§3.1). Therefore, even with a data budget of 100%, it does not select all data points, making the percentage does not reach 100%.

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

# A EXPERIMENTAL DETAILS

**Implementation Details**    All experiments were conducted using the AdamW optimizer. For most models, we set the learning rate to 2e-5, except for DeBERTaV3$_{Large}$, where we followed He et al. (2023) and used 1e-5. Additionally, we warmup the learning rate for the first 10% of training steps. For gradient-based pruning, the reference models were trained with LoRA, using a higher learning rate of 1e-4, $r = 64$, and $\alpha = 16$ following Ivison et al. (2023), and apply LoRA on all linear layers. We train all models for 15 epochs, For batch size, we used 16 for both WinoGrande and DialogSum, and 32 for CAD, to fit all experiments on a single NVIDIA A100-40GB GPU. We use maximum sequence lengths of 300, 128, and 512 tokens. For all experiments we use the same reference models as the main models for fair comparison.

For $k$-Means clustering in S2L, Prototypicality, and SemDedup, we use 100 clusters on CAD and DialogSum, and 200 clusters on WinoGrande, following the suggestions from Tirumala et al. (2023) to set the number of clusters to around the square root of the number of instances. Moreover, we compute gradients using the first five checkpoints for all experiments, and project them into a 1024-dimensional space using Park et al. (2023) (details see hyperparameter search).

**Evaluation Metrics**    We evaluated CAD and DynaHate using the macro F1 score, WinoGrande by accuracy, and DialogSum by ROUGE-1, ROUGE-2, and ROUGE-L (from HuggingFace Evaluate), following the original studies (Vidgen et al., 2021; Sakaguchi et al., 2021; Chen et al., 2021).

**Infrastructure**    All experiments were run on a single NVIDIA A100-40GB GPU using three random seeds. We used PyTorch 2.3, Transformers 4.42, and vLLM 0.5 for training and inference. Moreover, we use `bfloat16` on all experiments to improve efficiency.

**Hyperparameter Search**    We mainly searched for four hyperparameters: the number of training epochs, the number of clusters for $k$-Means clustering, the dimensionality of the projected gradients, and the checkpoints to use for gradient computation.

For the number of training epochs, we first perform a search of 3, 5, 7, and 10 epochs on all datasets and models, using three random seeds. We observe that models of different sizes share similar performance trends over epochs, with improvements continuing as the number of epochs increased. We therefore use the smaller models, i.e., DeBERTaV3$_{Base}$ and OPT-125M, and extend this search over 15, 20, and 25 epochs. Across all datasets, the best performance is achieved with 15 epochs.

For the number of clusters, we search over 2, 5, 10, 20, 50, 100, and 200 clusters for each dataset and model, using three random seeds. The results are highly consistent across cluster numbers. Following Tirumala et al. (2023), we use the square root of the dataset size as a guideline, settling on 100 clusters for CAD and DialogSum, and 200 for WinoGrande.

For gradients, we use smaller models (DeBERTaV3$_{Base}$ and OPT-125M) for hyper-parameter search, and only one random seed (0) to avoid the high costs of computing and projecting gradients. We compute the gradients for all 15 checkpoints, and project them into 1024, 2048, and 4096 dimensions. First, we observe that different dimensionality compute similar results, and thus choose 1024 for further experiments for efficiency. Second, we experimented with different strategies for selecting checkpoints, including the first three, the last three, the first five, the last five, and evenly spaced three and five checkpoints. Using the first checkpoints is the most consistent with using all checkpoints, with the first five yielding a minimum Spearman's rank correlation of 0.96. We therefore use the first five checkpoints for all experiments.

# B OVERVIEW OF DATA PRUNING METHOD

| Selection | Representations | | |
| --- | --- | --- | --- |
| | **Training dynamics** | **Hidden states** | **Gradients** |
| Max. diversity | SmallToLarge (Yang et al., 2024) | SemDedup (Abbas et al., 2023) | |
| Max. difficulty | Hard-to-learn (Swayamdipta et al., 2020; Jiang et al., 2021; İnce et al., 2023) | Prototypicality (Sorscher et al., 2022) SemDedup (Abbas et al., 2023) | Memorization (Feldman & Zhang, 2020) |
| Val. contribution | | | LESS (Xia et al., 2024) |

Table 1: Pruning methods from §3.1, categorized by their representations and selection objectives.

# C    ADDITIONAL RESULTS

## C.1    TOY EXAMPLE

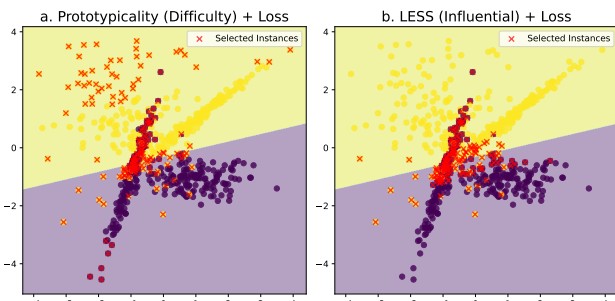

Figure 6: Interactions between data representations and selection algorithms. We generate 600 data points from a 2D Gaussian mixture model and compare different methods to select 30% (180) of the data points. The color represents the ground-truth label, and the red Xs are the selected data points.

## C.2    CONSISTENCY BETWEEN DATA PRUNING METHODS

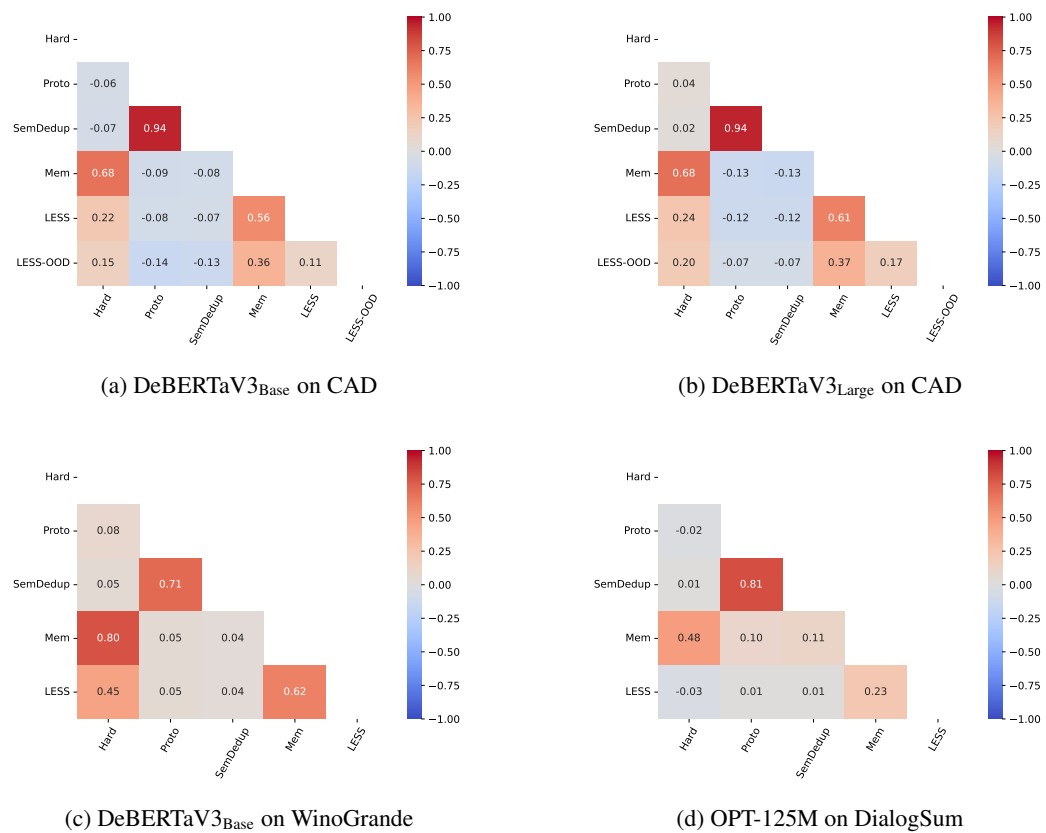

Figure 7: Spearman correlation of scores calculated by different methods (all methods select instances with the highest scores to retrain).

## C.3    PERFORMANCE UNDER DIFFERENT DATA BUDGETS

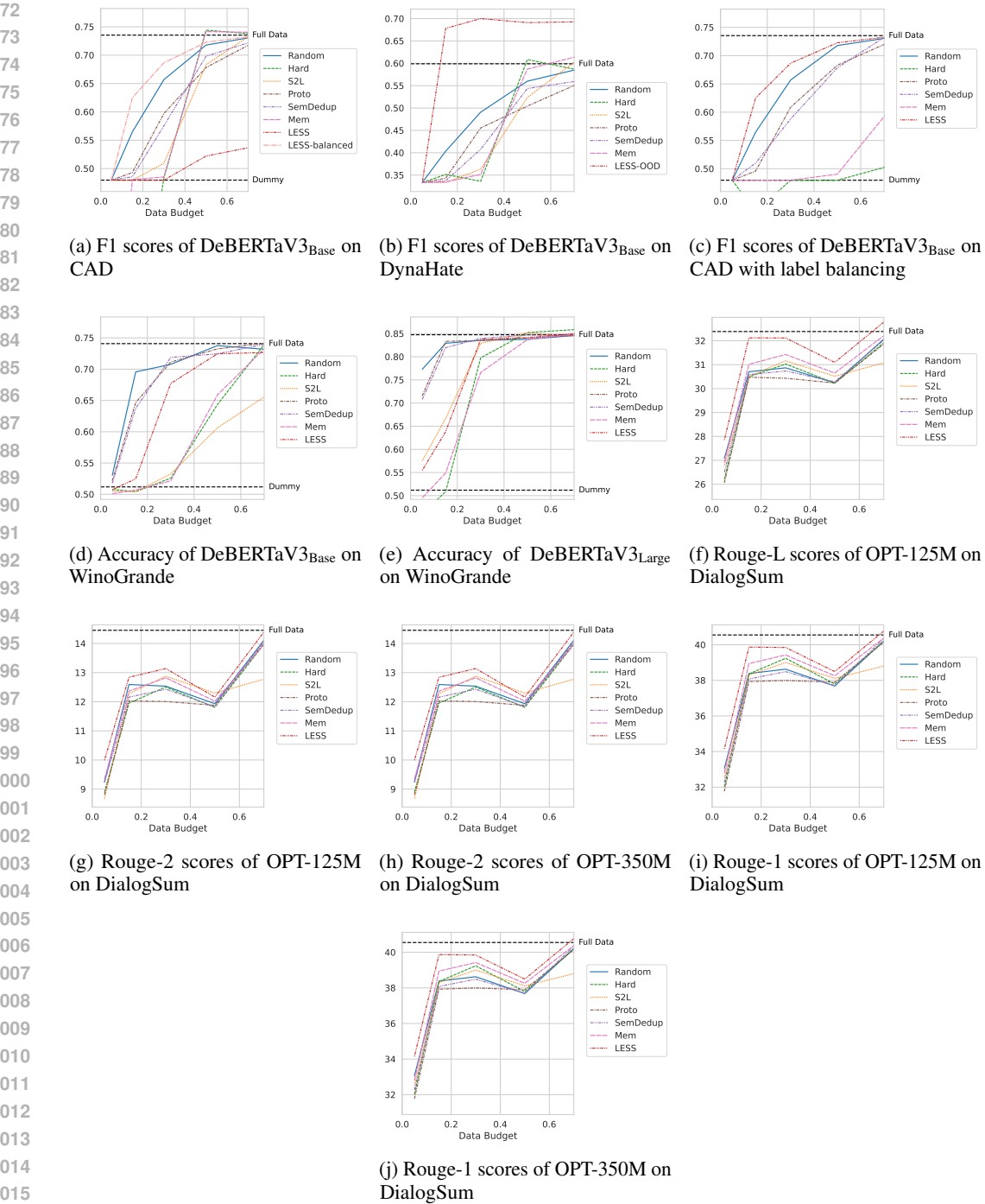

(a) F1 scores of DeBERTaV3$_{Base}$ on CAD

(b) F1 scores of DeBERTaV3$_{Base}$ on DynaHate

(c) F1 scores of DeBERTaV3$_{Base}$ on CAD with label balancing

(d) Accuracy of DeBERTaV3$_{Base}$ on WinoGrande

(e) Accuracy of DeBERTaV3$_{Large}$ on WinoGrande

(f) Rouge-L scores of OPT-125M on DialogSum

(g) Rouge-2 scores of OPT-125M on DialogSum

(h) Rouge-2 scores of OPT-350M on DialogSum

(i) Rouge-1 scores of OPT-125M on DialogSum

(j) Rouge-1 scores of OPT-350M on DialogSum

Figure 8: Model performance under different data budgets.

## C.4 ABLATION STUDY

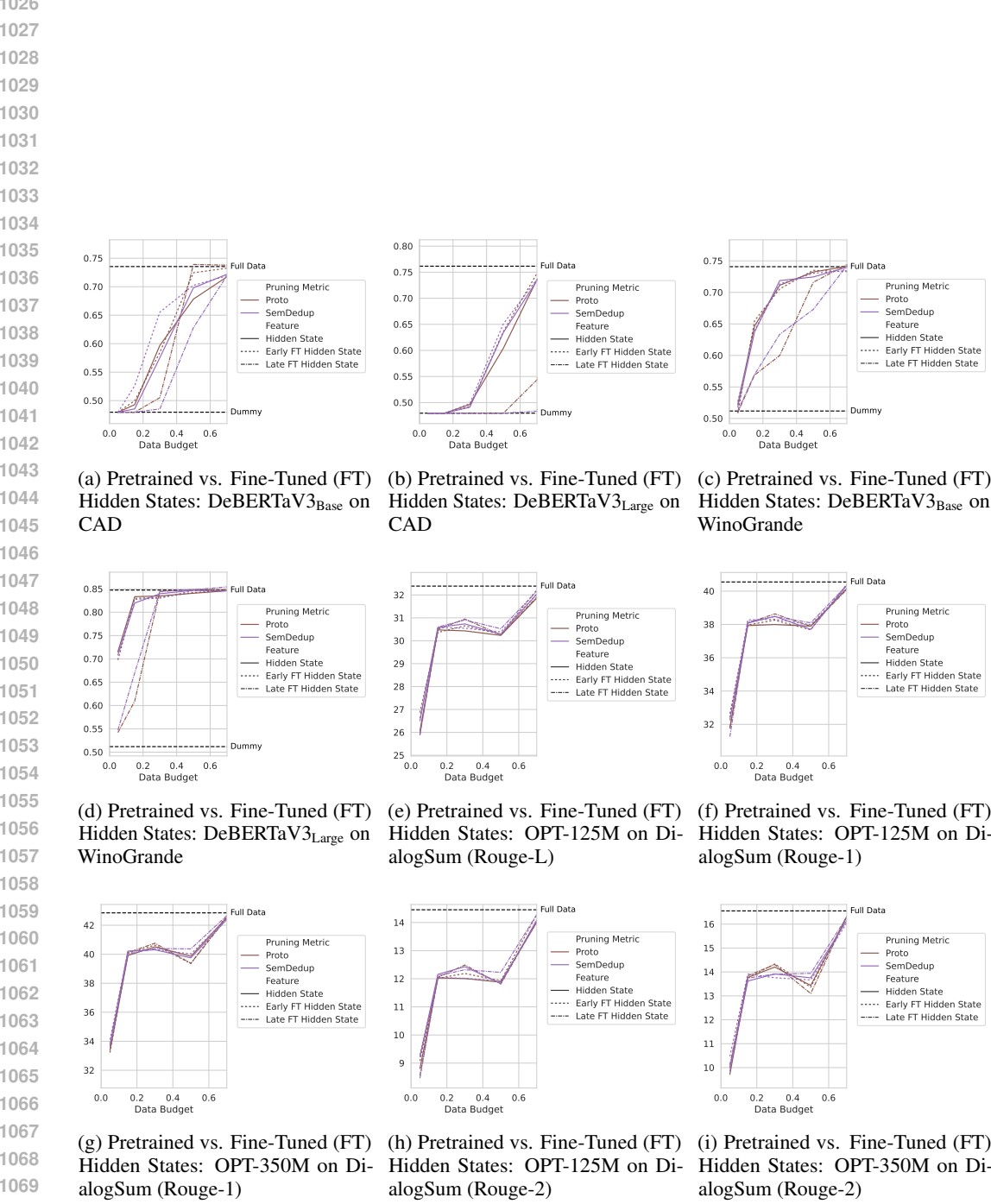

Figure 9: Ablation studies on pretrained vs. fine-tuned hidden states.

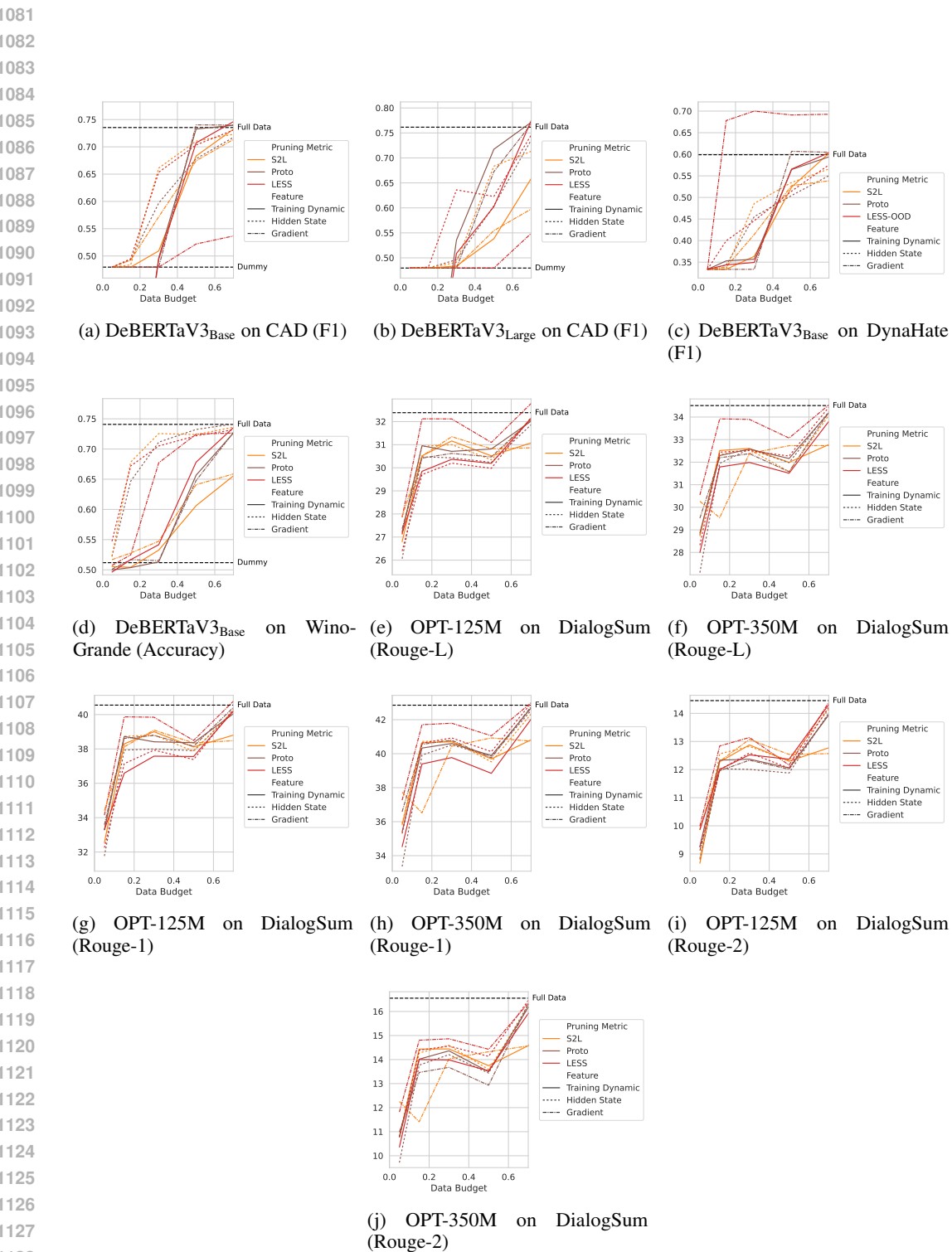

Figure 10: Ablation studies on using different data representations with the same selection algorithm.

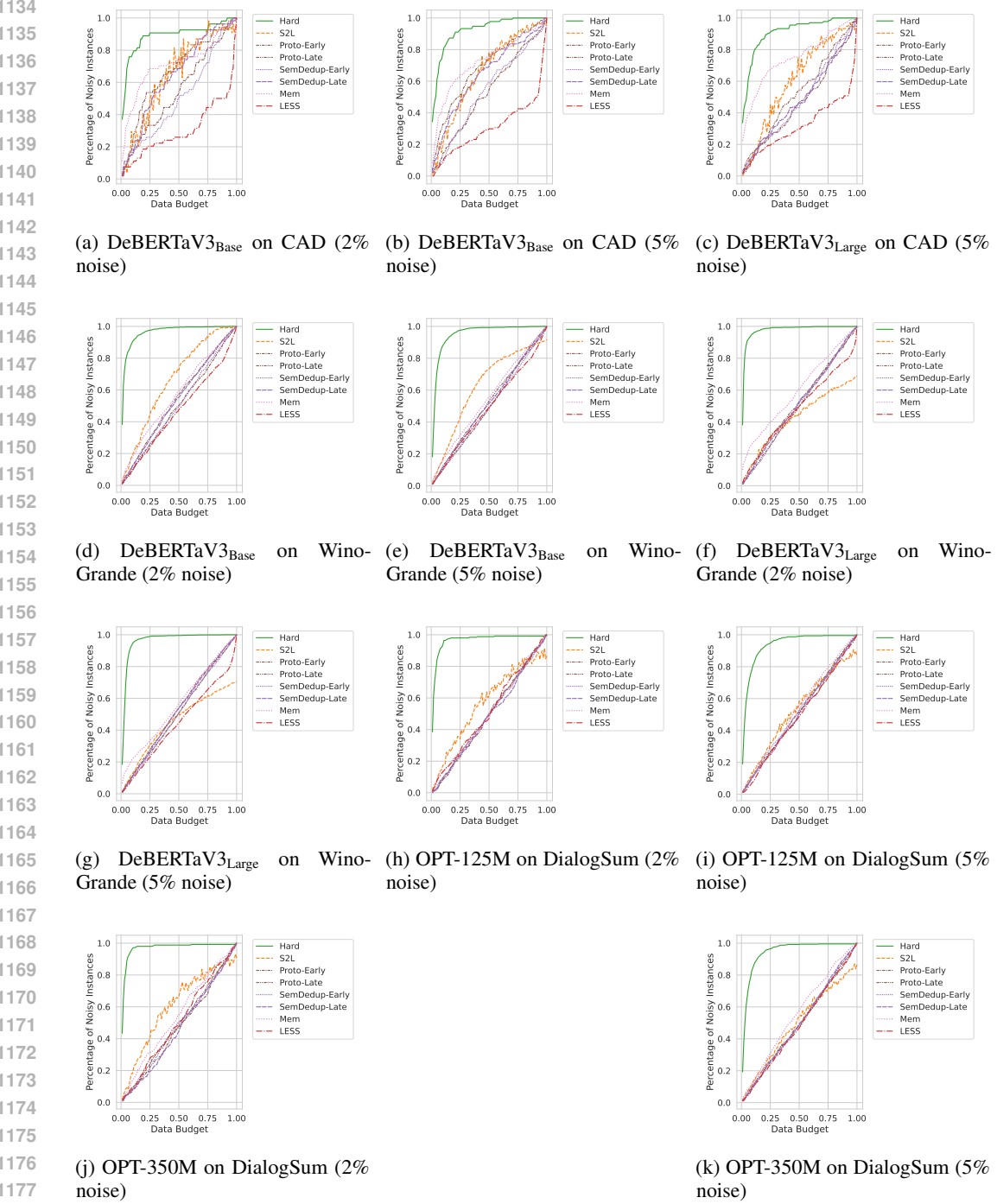

(a) DeBERTaV3$_{Base}$ on CAD (2% noise)

(b) DeBERTaV3$_{Base}$ on CAD (5% noise)

(c) DeBERTaV3$_{Large}$ on CAD (5% noise)

(d) DeBERTaV3$_{Base}$ on Wino-Grande (2% noise)

(e) DeBERTaV3$_{Base}$ on Wino-Grande (5% noise)

(f) DeBERTaV3$_{Large}$ on Wino-Grande (2% noise)

(g) DeBERTaV3$_{Large}$ on Wino-Grande (5% noise)

(h) OPT-125M on DialogSum (2% noise)

(i) OPT-125M on DialogSum (5% noise)

(j) OPT-350M on DialogSum (2% noise)

(k) OPT-350M on DialogSum (5% noise)

Figure 11: The percentage of selected noisy instances in all noisy instances for different models, datasets, and noise rates.

