# OpenReview forum: "Disentangling the Roles of Representation and Selection in Data Pruning (for Fine-Tuning)"
_ICLR.cc/2025/Conference — ICLR 2025 Conference Withdrawn Submission_

### Official Review · Reviewer_Lgd8 · 2024-10-21

**Soundness:** 3
**Presentation:** 2
**Contribution:** 2
**Rating:** 3
**Confidence:** 4

**Summary:**

This work is about data pruning methods—algorithms that score individual datapoints and retain small/moderate subsets to maximize model performance. The authors suggest that certain methods can be disentangled into two stages, 1) deriving a representation for each data point and 2) applying a scoring/selection mechanism based on those representations. The paper provides an overview of these methods, some analysis of the relationships between certain implementation choices, and then performs some empirical comparisons. The goal is to provide a deeper understanding of these methods and guidelines for future usage.

**Strengths:**

- The paper overviews a large number of related works, and provides a short and simple description of many of them
- It seems to introduce a previously unrecognized perspective in this line of work, that many methods can be disentangled into separate choices for their underlying representation and their scoring/selection rule
- The theory suggests some similarity between the different representation choices (hidden states, training dynamics and gradients)
- The experiments compare several existing methods, along with new methods combining different representation and scoring combinations that were not previously explored

**Weaknesses:**

One of the main points of the paper is disentangling the two main implementation choices for certain pruning methods (the representation and scoring rule). There are a couple aspects of this overview of related works that could be improved:

- At the beginning of the paper, it was difficult to tell what the authors meant by their second stage. The second paragraph includes the following text: "second, selecting instances based on these representations given a data budget (e.g., 30% of the training set), according to a selection algorithm." Perhaps you could provide a quick example, like the distance to cluster centroid idea, to clarify that the free parameter in this stage is not simply the amount of data to retain.
- The overview of methods at times reads like a laundry list with too little structure (mainly section 3.1). To emphasize that you're identifying consistent implementation choices within each method, it would help to introduce notation for each one (the representation and scoring approaches) and include tables showing the options demonstrated by these methods. Otherwise, you're putting too much burden on the reader to remember each method and perform this mapping themselves.
- Disentangling between the two stages would be especially interesting if you can identify new combinations of implementation choices that are promising new methods. I believe this was done to some extent in figure 1 (an experiment that does not measure performance), and a bit more in figure 4. Can you clarify whether any new useful algorithms were discovered, or whether more exploration of this type seems worthwhile? The paper did not seem as focused on deriving new and improved methods as I would have expected, and instead focused on describing differences.
- The paper ultimately gives the sense that many of the methods surveyed here basically don't work, or only work in specific settings (mainly the methods in section 2). Is there any way to focus the discussion on the methods that matter most, or provide some context to the reader about what works well or is popular in practice?

There are a couple remarks about related works that seem off:

- In lines 117-118, the authors mention Pruthi et al. as an example of using influence functions to estimate datapoint importance. This work does not describe itself as using influence functions (it compares to them), so is this a novel interpretation you're providing? If so, that may merit some explanation.
- In lines 125-127, the authors refer to Feldman & Zhang as an example of using gradients as a measure of self-influence. This work doesn't use gradients, it compares predictions after retraining with different datasets.

About the theory in section 3.2:

- One of the comparisons you make here is between $|p_\theta(y_i \mid x_i) - p_\theta(y_j \mid x_j)|$ and $||g_w(x_i, y_i) - g_w(x_j, y_j)||_2$. Based on the various methods you described previously, I'm not sure if either of these are used by any existing works? The second is at least related to gradient inner products (relevant to influence functions, TRAK, LESS), but the first quantity doesn't seem like it has any potential for effective data pruning. Why is it a useful point of comparison?
- Related to my request above for putting each method's implementation choices in a table: because that isn't currently in the paper, it's hard to keep track of which methods this section might apply to, even if indirectly. The summary says that the main takeaway is there are similarities between analyzing training dynamics, gradients and hidden states; that seems to imply certain existing methods are more similar than people realize, but I can't tell which methods those would be. Anchoring this subsection in specific methods seems important, otherwise it's a bit disconnected.

The conclusions from the simulated experiments in section 3.3 are not very surprising. They are the following: 1) methods with the same objective (e.g., finding difficult examples) select different datapoints depending on their representation/scoring (of course), 2) the same scoring rule can select different datapoints depending on which representation they use (naturally), but 3) sometimes not (for the sole case of the diversity-prioritizing method, which makes sense). These experiments are a bit more like sanity checks than providing new insights.

It would have been nice if the paper made generalizable claims about which methods work well, or perhaps components of methods that reliably work well (e.g., a representation that often works well with different scoring criteria). The closest we get to that is section 4, which compares a handful of methods on a few datasets. Although we get a sense of which methods score datapoints similarly (only a couple pairs of methods), and we see that a couple methods consistently don't work (those based on hidden states), the paper overall does not go very deep into the pros/cons of different implementation choices or attempt to make generalizable claims. Perhaps you could clarify what the field should take away from this newfound relationship between methods, particularly regarding the development of improved algorithms (since all of the methods tested here fail at some point in the experiments)?

A few other points about the experiments:

- In lines 385-386: these points about Spearman correlation are also true for Pearson correlation, are the results consistent across metrics?
- You included an extra point on the x-axis for LESS in fig 2a-b.
- The text does not describe how LESS and memorization produce relatively similar scores (fig 2a-b).
- Why do you include LESS-balanced in fig 2d when none of the others methods have their balanced counterparts here? It might also make more sense to put fig 2f next to fig 2d.
- Why do you use LESS-OOD in fig 2e? If this is different than the normal application of LESS here, perhaps you can show the normal usage as well?

**Questions:**

Several questions are mentioned in the weaknesses above.

---

### Official Review · Reviewer_UMtE · 2024-10-31

**Soundness:** 2
**Presentation:** 3
**Contribution:** 2
**Rating:** 3
**Confidence:** 4

**Summary:**

This paper investigates the roles of data representation and selection algorithms in data pruning for NLP fine-tuning tasks. The study divides existing data pruning methods into categories based on data representation (training dynamics, hidden states, and gradients) and selection objectives (maximizing difficulty, diversity, or validation performance). Through theoretical analysis, they show that gradients and prediction probabilities encode more information than hidden states as data representations. Through extensive experiments on both synthetic datasets and NLP datasets, the authors claim that data representations play a more significant role than the selection algorithms, largely affecting the samples selected. They discover that gradient-based representations achieve better performance under their single-task NLP fine-tuning scenarios.

**Strengths:**

1. The paper provides a broad review of current data pruning methods, considering various representations (training dynamics, hidden states, gradients) and selection objectives (difficulty, diversity, validation performance), which helps readers understand the current landscape of data pruning.
2. The paper offers a theoretical analysis of three data representations—hidden states, prediction probabilities, and gradients—revealing the signals captured and encoded in the similarities of these representations.
3. It conducts an extensive experimental comparison of data representations and selection algorithms, evaluating them on both synthetic and NLP datasets. The paper emphasizes that data representations tend to be more fundamental than selection algorithms in determining the quality of pruned data and gradient-based representations generally perform better. This provides practical guidance for data pruning for NLP task-specific fine-tuning.

**Weaknesses:**

1. The paper does not provide a systematic review of existing data pruning methods. The authors focus only on the contents of the methods (including data representations and selection algorithm), while overlooking the specific scenarios these methods are designed for. This narrow focus makes their attempt to disentangle representations and selection algorithms appear artificial. In practice, data pruning methods are designed as unified systems, with both components tailored for targeted scenarios. The authors’ approach of swapping representations and applying them in a single-task setting may diverge from the methods’ original intent, limiting the credibility of universal conclusions.
2. Although the authors note in the Limitation section that this study focuses on single-task fine-tuning, the experiments include methods that are designed for other scenarios. For instance, S2L is developed for domain-specific fine-tuning, and LESS targets downstream task-specific fine-tuning. Comparing these methods in a uniform task-specific setting disregards their intended contexts, creating an unfair comparison that undermines the validity of the findings regarding the superiority of data representations or methods.
3. Some prior work is misinterpreted. In section 2.1, methods are characterized into maximizing data difficulty as the selection objective. However, the mentioned work When Less is More by Marion et al. finds that data of moderate difficulty is most beneficial, which contradicts the paper’s claim. In section 3.3, the authors list S2L as “representation-agnostic”. However, the data representation (training trajectories) used in this algorithm is specially designed to address domain-specific fine-tuning challenges where hidden states may be less effective.
4. While the empirical results are valuable, they do not provide significant new insights. In 3.3 and 4.2, the authors “find” that different selection methods with the same difficulty objective select different samples. However, as “difficulty” is not a fixed data property but a human-defined measurement that differs across methods, this variation is unsurprising. Moreover, LESS, which maximizes validation performance, is naturally expected to have better performance under the authors’ single-task fine-tuning setting, but this may not generalize to other unexplored settings.
5. The presented work focuses solely on evaluating existing approaches. No new data pruning methods or frameworks are proposed, which limits its originality and methodological contribution.

**Questions:**

1. While the paper focuses on task-specific fine-tuning and excludes general instruction fine-tuning, could the authors discuss whether the findings could extend to broader contexts? Or are the findings limited to specific tasks?
2. Methods like LESS and S2L were originally applied to billion-parameter models, but the current experiments use models with only up to 350M parameters. Have the authors considered testing on larger models, such as LLaMA-7B or at least Pythia-1B, to improve the study’s credibility and applicability?

---

### Official Review · Reviewer_CmBp · 2024-11-02

**Soundness:** 1
**Presentation:** 3
**Contribution:** 2
**Rating:** 3
**Confidence:** 4

**Summary:**

The authors conduct a review of data pruning methods. They identify two key components of these methods: the representation of the data, and the pruning strategy applied on top of this representation. The representations are divided into three broad categories: gradient based, hidden state based, and training dynamics based. The pruning strategies also have three broad categories based on what they seek to maximize: diversity, difficulty, or performance (on a validation set). They conduct experiments using these pruning strategies on synthetic 2D Gaussian mixture datasets to show the impact different combinations of representation/pruning strategy have on the final selection. Finally, they test the different pruning strategies on several real-world NLP tasks and conclude that gradient-based methods tend to have the best performance, albeit at the cost of computational efficiency.

**Strengths:**

Reproducibility studies or extensive comparisons of existing techniques have very high practical value for the ML community. Especially as datasets and models continue to grow in size, strategies for speeding up training, removing low quality data, and reducing storage costs (such as data pruning) will be increasingly important. Thus, this topic is highly relevant to the ICLR community.

The authors summarize and explain the methods under study very clearly. The related work section is also extensive and clearly written.

The paper also offers a key insight into why LESS, which was generally the most performant pruning method, failed on the CAD dataset due to a label imbalance which was exacerbated by the method (lines 448-452). They then propose a simple alteration (label balancing) which restores the performance of LESS.

**Weaknesses:**

There is a flaw in the reasoning for the theoretical analysis on lines 243-244. The authors claim that because the sigmoid function is smooth and monotonically increasing, when $|\sigma(x) - \sigma(y)|$ is small it should also be the case that $|x-y|$ is small, but this is not true. In fact, since the sigmoid flattens out at $\pm\infty$, the difference in sigmoids can be made arbitrarily small while the difference in arguments is arbitrarily large. (E.g., $|\sigma(x) - \sigma(x^2)|\to 0$ but $|x-x^2|\to\infty$ as $x\to\infty$.) I don't see any easy way to fix this flaw in the reasoning.

The motivation for the theoretical analysis conducted in Section 3.2 is not clear, as the quantities studied (difference in correct output probabilities or gradients) aren't actually used by any of the pruning methods. The connection between the theoretical analysis and the pruning methods under study should be made more explicit.

The value of some of the main conclusions is also not clear. For instance, on line 309, the authors emphasize that "even when data pruning methods have the same objective, the representations and selection algorithms used can result in drastically different subset selections." This is not necessarily surprising, as the data pruning methods do not *literally* have the same objective, they just fall into the same category defined by the authors of the present paper. An equally plausible explanation for this phenomenon is that the categories of objectives defined in the present paper do not meaningfully separate pruning objectives.

Finally, while reproducibility or comparison studies of existing work *can* be quite valuable, they should be exhaustively thorough in order to merit publication at a top venue like ICLR. There were many other methods listed in the related work section which were not tested, and it is not clear if the conclusions drawn in the paper should extend to the many other methods available. In order to provide convincing evidence that the decomposition into representation + selection strategy and the classification of representations/selection strategies defined by the authors are meaningful, other related methods should be tested to verify that the qualitative claims made by the paper are generally applicable.

**Questions:**

What exactly is the meaning of "training instances that are difficult for models to fit often contain fewer regularities" (lines 162-163)?

---

### Official Review · Reviewer_FAXu · 2024-11-04

**Soundness:** 2
**Presentation:** 2
**Contribution:** 2
**Rating:** 3
**Confidence:** 5

**Summary:**

This paper investigates the important research problem of understanding the roles of representation and selection strategy in data prunning problems. Data pruning is often conducted in some representation space, but the choice of representation is often different for different methods or in different use cases. The respective role for the choice of representation and the data prunning strategy has long been unclear. This work aims to contribute a more systematic study towards this problem.

The paper first reviews and organizes a list of commonly used data selection/prunning methods and categorizes their representations spaces into "Training dynamics", "Hidden states", and "Gradients". Then, the paper analyzes different representations and their interaction with data selection strategies with derivations and simulations on stylized models. Finally, the paper conducts a number of experiments on 3 NLP fine-tuning tasks with pre-trained models, DeBERTaV3 and OPT, and draws a number of conclusions on the findings. The paper also conducts ablation studies on fine-tuned embeddings and additional tasks, etc.

**Strengths:**

The paper is very well-positioned. The problem being investigated, understanding the respective role for representation and data selection strategies, is crucial and highly relevant. Contributions from this angle are very much anticipated.

This paper serves as a nice, compact study. It studies a meaningful and timely problem and is self-contained and reasonably structured. The data pruning/selection methods implemented in this work are diverse and representative, covering a variety of seminal works. The literature review is also favorable.

The paper has a nice combination of theoretical analysis, synthetic simulations, and empirical studies on commonly used benchmarks. The findings and conclusions are in the right direction.

**Weaknesses:**

The manuscript being reviewed contains a number of inconsistencies and ambiguities in multiple places. Some key concepts are not clearly distinguished from each other and are seen used interchangeably. Also, some modifications to the baseline methods seems go beyond the scope of the original works, which essentially becomes new methods rather than the ones with the original names. Experiments are confined to fine-tuning for NLP tasks, which are not  what most of these methods are proposed for. Experiments are all in small scales. Theoretical analysis does not provide many in-depth insights. The conclusions and findings are in the right direction and does not tell much beyond the prior hypotheses.

Some detailed feedback.

1. The title of the paper says "for fine-tuning", but it is not referred to or discussed about throughout the paper except the experiments are all fine-tuning tasks.

Data pruning for fine-tuning could potentially be a quite different problem. The model already has prior knowledge, so while selecting data, one may want to avoid samples the model is already doing well, (or not). For example, in instruction-tuning tasks for LLMs, following [Lima: Less is more for alignment. C Zhou et al.], a wealth of works have been proposed to achieve comprable performance with a small fraction of instruction samples.

None of the baseline methods implemented in this work was proposed for such use cases. Rather, these methods focus on the case of training from scratch. Mentioning fine-tuning in the title or confining the experiments to such tasks disconnects with the main content of this paper.

Besides, for fine-tuning, the practical challenge is often the lack of high-quality data rather than the computation cost. If one considers large-scale fine-tuning, which may become continual learning. It may also be a problem of its own flavor.  Ref: [Scaling laws for transfer. D Hernandez et al.]


2. This paper uses the notions of data pruning and data selection problems interchangeably. There are several important distinctions.
Data pruning often refers to the case with an (over)abundance of data where the model can achieve comparable performance by training discarding redundant training data. The goal is to retain the model's original performance while removing as many data as possible. It is typically done in a one-shot manner. Except for Memorization, which was proposed for understanding data influence, all other baseline methods implemented in this work belong to this type. That's why they often prioritize hard samples or remove duplicate samples since those samples provide little marginal contribution at a large-data regime.

On the other hand, data selection for machine learning studies the general meta-learning problem of how to select training data to optimize certain objectives for the resulting model, which could be performance/efficiency/fairness, etc. Muti-round data selection methods are often categorized as active learning, which has a rich field of literature. If the goal is to achieve the best possible performance with a small data budget (such as 20% of the original dataset), the task is often referred to as coreset selection.

Further, noisy data selection is a diagnostic problem and not the main consideration for data pruning. This kind of problems are often dealt with data influence or data valuation methods, which try to understand/quantify individual data point's contribution to the set objective (e.g., model performance).

Consider
- Clearly define these terms early in the paper
- Consistently use the appropriate term throughout
- Discuss how their findings may differ for data pruning vs data selection tasks

3. The paper could benefit from restructuring some of the sections. For example, Section 3.2, the information is not very straightforward.  The style is a mix of elaboration and derivations. If it is intended to be theoretical analysis, structuring it in Theorem-Remark style may substantially improve both its rigor and clarity.

Consider presenting the key theoretical results as formal theorems or propositions, followed by explanations and implications. This would help separate the main analytical insights from the supporting details.

4. Main conclusions are overly ambiguous. Conclusion 1: "data pruning methods may not be effective". This does not tell much more than intuitions–most methods may not always work. A crucial research question is "when these methods are effective and when they are not", which may reveal intrinsic patterns which are previously unknown and guide the research for future improvements. Similarly for the conclusion, "representations are more important".  Why is it more important? Is it always the case?

Consider
- Provide more specific conditions under which data pruning methods are or are not effective
- Quantify the effectiveness (or lack thereof) of different methods in various scenarios
- Discuss the implications of these findings for future research directions in data pruning

**Questions:**

1. This work makes it default to use "representations from the model that we are training. This allows us to analyze signals that directly reflect its learning behavior." This is a very strong assumption, basically requiring the embedding to be perfectly aligned with the target case/model/data. This may not always be possible in practice, especially when data selection needs to done in one-round rather than in online/active-learning fashion. The mismatch between embedding space and target tasks could significantly affect the effectiveness of data selection pipelines. Given that the paper aims to study this problem as its main focus, it is not very reasonable to skip this discussion here.

2. The paper considers "memorization" as a "gradient"-style method and argues "Using the TracIn influence function, this self-influence score can be estimated as the gradient norm." This is not proposed in the original paper [Feldman and Zhang, 2020]. Is this originally proposed in this work? It is not an established practice to approximate memorization scores with first-order gradients. As mentioned in the training dynamics part of the paper, this actually approximates only the final step gradients and may or may not reflect what the model has picked up from this sample during training. Please clarify if this is a novel interpretation or adaptation of the original method. If it is novel, justify why this approximation is valid and discuss its limitations; if it is not novel, provide proper citations for this interpretation.

3. The experiments on "fine-tuning hidden states" may not tell the full story. For example, for the task of hateful speech recognition with pre-trained BERT models, if using vanilla BERT models which have no knowledge of the target tasks, its embedding space distance may not be relevant to whether a sample is considered "hateful" or not. But after training the model on this classification task, its final-layer embedding will push samples with different labels to different clusters which are often linearly separable. Conducting data selection on such embedding space is likely to yield very different results than using task-agnostic embeddings.

---

### Author Response · Authors · 2024-11-22
**Thank You**

We sincerely appreciate the thoughtful and detailed feedback provided by the reviewers. We will use their insights to improve our work and prepare a stronger version in the future. Thank you for your time and effort.

---

### Note · Authors · 2024-11-22

I have read and agree with the venue's withdrawal policy on behalf of myself and my co-authors.